# Pathological Studies and Postmortem Computed Tomography of Dolphins with Meningoencephalomyelitis and Osteoarthritis Caused by *Brucella ceti*

Andrés Granados-Zapata [1], María José Robles-Malagamba [2], Rocío González-Barrientos [3], Brian Chin-Wing Kot [2,4], Elías Barquero-Calvo [5], Minor Cordero-Chavaría [6], Marcela Suárez-Esquivel [5], Caterina Guzmán-Verri [5], Jose David Palacios-Alfaro [7], Connie Tien-Sung [8], Edgardo Moreno [5,*] and Gabriela Hernández-Mora [6,*]

[1] Escuela de Medicina Veterinaria, Universidad Nacional, Heredia 40104, Costa Rica; andres_g136@hotmail.com
[2] Department of Infectious Diseases and Public Health, Jockey Club College of Veterinary Medicine and Life Sciences, City University of Hong Kong, 83 Tat Chee Avenue, Kowloon, Hong Kong; mj.robles90@my.cityu.edu.hk (M.J.R.-M.); briankot@cityu.edu or briankot@cityu.edu.hk (B.C.-W.K.)
[3] Section of Anatomic Pathology, Department of Biomedical Sciences, College of Veterinary Medicine, Cornell University, Ithaca, NY 14853, USA; cristyrocio@gmail.com
[4] Centre for Applied One Health Research and Policy Advice, City University of Hong Kong, 83 Tat Chee Avenue, Kowloon, Hong Kong
[5] Programa de Investigación en Enfermedades Tropicales, Escuela de Medicina Veterinaria, Universidad Nacional, Heredia 40104, Costa Rica; elias.barquero.calvo@una.ac.cr (E.B.-C.); marcela.suarez.esquivel@una.cr (M.S.-E.); caterina.guzman.verri@una.cr (C.G.-V.)
[6] Servicio Nacional de Salud Animal (SENASA), Ministerio de Agricultura y Ganadería, Heredia 40104, Costa Rica; minor.cordero.c@senasa.go.cr
[7] San Pablo de Heredia, Santiago, Heredia 40503, Costa Rica; pala1611@gmail.com
[8] Veterinaria Vicovet, San José 11502, Costa Rica; connie22ts@gmail.com
* Correspondence: emoreno@una.cr (E.M.); gabriela.hernandez.m@senasa.go.cr (G.H.-M.); Tel.: +506-25871837 (G.H.-M.)

**Abstract:** Cetacean neurobrucellosis is a common cause of strandings in Costa Rica diagnosed by serology, bacteriology, and histopathology. Pathological studies were performed on 18 dolphins. Twelve were scanned by postmortem computed tomography (PMCT) as a complementary tool for describing neurobrucellosis and osteoarticular alterations associated with *Brucella ceti* infections. The central nervous system (CNS) and the skeleton of five dolphins not infected with *B ceti* did not reveal alterations by PMCT scanning. Seven *Brucella*-infected dolphins showed at least bilateral ventriculomegaly associated with hydrocephalus and accumulation in CSF in the lateral ventricles. We performed semiquantitative grading of the inflammatory process in the different areas of the CNS and evaluated the thickness of the cellular infiltrate in the meninges and the perivascular cuffs. The results for the severity grading were graphed to provide an injury profile associated with each area of the CNS. Age is not a decisive factor regarding neurobrucellosis presentation. The severity of ventriculomegaly by PMCT does not directly correlate with the severity of the inflammatory index determined by histopathological parameters of the brain cortex and other CNS regions, suggesting that these processes, although linked, are multifactorial and need further characterization and validation to establish better cutoffs on the PMCT.

**Keywords:** neurobrucellosis; ventriculomegaly; osteoarthritis; Costa Rica; dolphins; *Brucella ceti*

## 1. Introduction

The extant cetacean fauna in the oceans consists of 92 species, 30 of which are present in the Costa Rican marine domains: 22 have been confirmed in the Pacific Ocean and 9 in the Caribbean Sea [1,2]. These diverse and abundant cetacean populations indicate the well-being of the Costa Rican oceans and gauge the magnitude at which the marine

resources are protected. Furthermore, cetaceans attract tourists worldwide and represent a source of income for coastal towns in a sustainable tourism framework.

As in other latitudes, cetacean species in Costa Rica can be exposed to several infectious diseases such as those caused by viruses, protozoa, fungal, and bacterial organisms that affect various tissues. Those affecting the central nervous system (CNS) are particularly interesting since they commonly cause stranding and host death [3–6]. Among the bacterial infections, brucellosis, due to *Brucella ceti*, is a common disease of these marine mammals [4].

*B. ceti* infections in cetaceans produce a complex syndrome characterized by lesions in different organs such as the spleen, liver, heart, bone marrow, lungs, bones, brain, mammary gland, and testes. In pregnant animals, the bacterium induces abortion due to invasion of the placenta and fetal tissues [7–9]. Cetacean neurobrucellosis is a common cause of strandings since it causes disorientation, uncoordinated lateral swimming, buoyant disturbances, and death [4,7,9,10].

Although some brucellosis human cases caused by *Brucella* display genotypes similar to those bacteria isolated in cetaceans [9,11], there is no direct evidence of *B. ceti* transmission from these animals to humans. However, *Brucella* organisms are pathogens of mammals and a relevant source of zoonotic disease; therefore, stranded cetaceans in direct contact with domestic animals and humans are a zoonotic risk [4,8].

On the Costa Rican coasts, *B. ceti* infected *Delphinidae* strand alive, showing fair to mildly decreased body conditions with no gross lesions, except abrasions caused by the stranding event. Before death, stranded dolphins frequently display opisthotonus, tremors, and tonic–clonic seizures [4,7]. The infected animal's blood and cerebrospinal fluid (CSF) show antibodies against smooth brucellae, and the CSF is cloudy with severe pleocytosis of lymphocytes, plasma cells, and macrophages, some with intracellular bacteria [4]. After bacteriological testing of the CSF, CNS, and central organs, most serologically positive-stranded cetaceans render *B. ceti* sequence types 26 (ST26) genotype P1 (striped dolphins, *Stenella coeruleoalba*), or ST27 genotype H (e.g., dwarf sperm whale *Kogia sima*) [4,7,9,12].

CNS gross evaluation of cetaceans with neurobrucellosis reveals mildly thickened and opaque meninges with hyperemic blood vessels and, in some cases, internal hydrocephalus with enlargement of the lateral ventricles due to CSF accumulation. Histologically, diffuse, chronic, lymphoplasmacytic, and histiocytic meningoencephalomyelitis is a characteristic sign of neurobrucellosis [4,7,13]. The inflammatory process mainly affects the leptomeninges, the subependymal neuroparenchyma (periventriculitis), and forms thick perivascular cuffs, which generally become more severe in the spinal cord and the brainstem medulla oblongata and cerebellum and less prominent in the cerebrum [4,7,13]. In a few cases, *Brucella* organisms within cells of the perivascular cuffs and glial foci have been revealed by immunohistochemistry [13].

Similar to humans, cetacean brucellosis is also associated with osteoarticular complications, mainly in the vertebral column and atlantooccipital joint [10,14,15]. Spondylitis and discospondylitis with bacterial isolation are the most frequent complications of brucellar spinal involvement, which generally reveal a long-lasting infection. However, unlike humans, who have several synovial joints adapted for movement and weight-bearing stress, the joints of cetaceans are not adapted to weight but are devoted to the contraction of adjacent muscles. In dolphins, the atlantooccipital joint is one of a few synovial joints present and therefore affected by *B. ceti* infections [14]. However, reports of associated bone lesions in cetaceans are scarce compared to frequent osteoarticular complications observed in human brucellosis. In one report, *Brucella* species were isolated from the atlantooccipital joint of an Atlantic white-sided dolphin (*Lagenorhynchus acutus*), resulting in a partial to complete ankylosis with total loss of normal joint movement. These changes were associated with severe bone remodeling of the occipital condyles and the atlas with complete absence of all articular cartilage and smooth subchondral bone, in addition to remodeling of the margin of the foramen magnum [14]. These pathologies are relevant

because osteoarticular lesions may hamper the swimming performance of the cetaceans and affect their survival.

Noninvasive imaging techniques such as computed tomography (CT) and magnetic resonance imaging (MRI) are commonly used to diagnose neurobrucellosis and associated osteoarticular lesions in humans [16,17]. Although CT and MRI have been used in dolphins presumptively infected with *Brucella* organisms [18,19], there are no peer-reviewed documents on these subjects. Postmortem CT (PMCT) and postmortem MRI (PMMRI) are noninvasive techniques that create volumetric image datasets, while two- and three-dimensional volumetric reconstruction and rendering are performed with advanced visualization technology (e.g., multiplanar reconstruction and direct volume rendering). Imaging findings on cetaceans´ head, neck, heart, vascular system, thorax, abdomen, vertebral column, and pectoral fins have been later confirmed by necropsy examination [20–22].

Since the procedure is fast, performing PMCT on stranded dolphins is particularly interesting because it allows an objective visualization and recapitulation of postmortem findings before conventional necropsy [20–22]. Here we report the PMCT scans of the CNS and axial skeleton scans of stranded dolphins in Costa Rica infected with *B. ceti* and compare the findings with conventional pathology exploration.

## 2. Materials and Methods

### 2.1. Dolphins

Seventeen stranded dolphins under study, including *S. coeruleoalba*, *Stenella longirostris*, *Tursiops truncatus*, *Lagenodelphis hosei*, and *Delphinus delphis*, were collected in the Pacific and one in the Caribbean littorals of Costa Rica (Table 1). Upon stranding, dolphins were examined in situ by a veterinarian, following the protocols for sampling as part of the National Brucellosis Control Program and Wildlife Program of the Costa Rican National Animal Health Service (SENASA). Most animals were stranded alive but died within the following 1–2 h on the beach. Dead dolphins were transported on ice within hermetically sealed bags to the National Health Service laboratories within the next 1–3 h for gross examination, bacteriological sampling of blood, CSF and secretions, and serological studies. Upon arrival, CT scans followed by necropsy were performed on two dolphins. Six other dolphins were subjected to necropsy upon arrival (Table 1). Ten dolphins were immediately frozen horizontally at −20 °C within sealed bags and later analyzed by CT scanning reconstruction, followed by necropsy.

**Table 1.** Characteristics of the dolphins included in this study [a].

| Code | Species | Age | Sex | Stranding | Ocean | PMCT Days [b] | RBT/ cELISA | *B. ceti* [c] |
|---|---|---|---|---|---|---|---|---|
| CR 22-17 | *Stenella longirostris* | C | M | 2017 | P | 10 | Negative | Negative |
| CR 03-20 | *Stenella coeruleoalba* | A | M | 2020 | P | 0 | Negative | Negative |
| CR 04-20 | *Stenella coeruleoalba* | A | F | 2020 | P | 3 | Negative | Negative |
| CR 05-20 | *Stenella coeruleoalba* | C | F | 2020 | P | 3 | Negative | Negative |
| CR 01-21 | *Tursiops truncatus* | A | F | 2021 | P | 0 | Negative | Negative |
| CR 09-20 | *Lagenodelphis hosei* | A | M | 2020 | Cb | 20 | Positive | Negative |
| CR 03-18 | *Stenella coeruleoalba* | J | M | 2018 | P | 8 | Positive | Positive |
| CR 10-20 | *Stenella coeruleoalba* | A | M | 2020 | P | 3 | Positive | Positive |
| CR 05-19 | *Stenella coeruleoalba* | A | F | 2019 | P | 1 | Positive | Positive |
| CR 07-19 | *Stenella coeruleoalba* | A | F | 2019 | P | 2 | Positive | Positive |
| CR 08-20 | *Stenella coeruleoalba* | A | M | 2020 | P | 24 | Positive | Positive |
| CR 01-18 | *Stenella coeruleoalba* | C | F | 2018 | P | 2 | Positive | Positive |
| CR 08-17 | *Delphinus delphis* | A | M | 2017 | P | ND | Positive | Positive |
| CR 15-18 | *Stenella coeruleoalba* | A | F | 2018 | P | ND | Positive | Positive |
| CR 20-18 | *Stenella coeruleoalba* | A | M | 2018 | P | ND | Positive | Positive |
| CR 06-17 | *Stenella coeruleoalba* | A | M | 2017 | P | ND | Positive | Positive |
| CR 13-17 | *Stenella coeruleoalba* | A | M | 2017 | P | ND | Positive | Positive |
| CR 16-17 | *Stenella coeruleoalba* | A | M | 2017 | P | ND | Positive | Positive |

[a] A, adult; J, juvenile; C, calf; M, male; F, female; P, Pacific; Cb, Caribbean; PMCT, postmortem computed tomography, RBT, Rose Bengal test. [b] Time-lapse in days between arrival to the laboratory and PMCT. In dolphins CR 03-30 and CR 01-21, the PMCT was performed upon arrival. In all other dolphins, the PMCT was performed frozen at −20 °C. ND, no PMCT was performed in these animals, and necropsy was performed upon arrival. [c] Isolation of *Brucella ceti* dolphin type, ST26. Differential diagnoses of toxoplasmosis and morbillivirus were evaluated as described previously [7,10]. Codes in bold correspond to the entries in SENASA databases.

### 2.2. Postmortem Computed Tomography (PMCT)

PMCT was performed on 12 dolphins. Five CT scans were carried out from 2017 to 2019 and seven from 2020 to 2021. Dolphins were scanned in a dual-source Siemens SOMATOM Scope® (Malvern, PA, USA), 16-slice CT system, a device that supports a body weight up to 200 kg and a length of 2 m. Scanning was performed at 130 kV, from 23–25 mA, with slices of 1.50 mm thickness. The scanning field of view varied from 334–500 mm. Scan parameters (B41s, B30s, H30s, H47s, and U90s) for soft and bone tissues were handled through Syngo.via (Simens, Berlin, Germany) integrated imaging software, which selects the pitch value for a given coverage scan time while retaining the slice thickness and image integrity. The scanned datasets were rebuilt using the 3D multiplanar reconstruction tool in commercial-free DICOM Viewer (RadiAnt™, Poznań, Poland) to reconstruct images in arbitrary planes [21,22]. In addition, the 3D volume rendering tool provided by RadiAnt DICOM Viewer (Poznań, Poland.) was also used to observe large volumes of data in the three-dimensional space. These tools were used to reconstruct images in arbitrary planes (oblique, transverse, dorsal, and sagittal). The interpretation of CT scans was partially performed at the Aquatic Animal Virtopsy Laboratory at the City University of Hong Kong.

### 2.3. Necropsy, Sample Collection, and Histopathology

As previously described [7], necropsies were carried out on 18 stranded dolphins following the National Veterinary Services Laboratory guidelines of the USDA [23]. Tissue samples were obtained from the spleen, liver, lymph nodes, lungs, reproductive system, kidneys, mammary gland, brain, and bones. The brain and cerebellum of each dolphin were removed from the skull and examined macroscopically in detail. The brain's dissection was performed carefully to collect the CSF from the ventricles. The obtained CSF volume was estimated in milliliters. Next, the cellularity of the CSF was studied, and tissue samples were obtained from the spinal cord, medulla oblongata, pons, caudal colliculus, rostral colliculus, thalamus, basal ganglia, cerebellum, occipital cortex, parietal cortex, and frontal cortex to perform histopathological studies and bacterial isolation, as previously reported [4,13]. For semiquantitative grading of the inflammatory process in the different areas of the CNS, the thickness of the cellular infiltrate in the meninges, and perivascular cuffs were evaluated in the histological sections. Based on these observations, the severity of the inflammatory infiltrate was scaled as "0" none (0 layers), "1" mild (1–3 layers), "2" moderate (4–7 layers), and "3" severe (>7 layers). The results for the severity grading were graphed to provide an injury profile associated with each area of the CNS. Other lesions, such as hydrocephalus associated with neurobrucellosis, were identified during the necropsy and classified according to the amount of CSF collected. Differential diagnoses of other infections causing neurological disorders with mononuclear inflammation in cetaceans, such as toxoplasmosis and morbillivirus, were evaluated as previously described [7,10].

*2.4. Detection of Antibodies and Bacterial Isolation*

The blood sera and CSF of dolphins were tested for the presence of antibodies against smooth *Brucella* lipopolysaccharide (Br-LPS) by Rose Bengal agglutination test, indirect ELISA (iELISA), and competitive ELISA (cELISA) as previously reported [24,25]. Bacterial isolation was performed from tissues sampled at necropsy following previous protocols [26]. Briefly, non-selective and selective media, including blood agar, Farrel, and CITA medium, were used for bacterial isolation [26]. Bacterial cultures were incubated in a 10% $CO_2$ atmosphere at 37 °C for at least two weeks. The selected bacterial colonies were tested for oxidase and urease, $H_2S$ production, citrate utilization, nitrate reduction, growth in the presence of thionin (20 µg/mL), basic fuchsine (20 µg/mL), and uptake of crystal violet as described elsewhere [27]. In addition, Gram staining and serotyping were performed. Genotyping was carried out in all isolated bacteria as previously described following Bruce-ladder multiplex PCR, fingerprint profiles of insertion sequences IS711, multilocus sequence typing (MLST) to define the sequence type (ST) profile, multiple loci variable number of tandem repeats analysis (MLVA), and whole-genome sequencing (WGS) [12].

## 3. Results

*3.1. Dolphins*

The age, sex, and species of the 18 dolphins included in this study are presented in Table 1. Fourteen individuals belonged to the species of striped dolphins (*Stenella coeruleoalba*), and the rest to a spinner dolphin (*S. longirostris*), a bottlenose dolphin (*Tursiops truncatus*), Fraser's dolphin (*Lagenodelphis hosei*), and short-beaked common dolphin (*Delphinus delphis*). Except for Fraser's dolphin, which was stranded on the Caribbean coast, all other dolphins were stranded along the Pacific coastline.

*3.2. PMCT, Necropsy, Histopathology, Serology, and Bacterial Isolation*

PMCT was performed on 12 dolphins. The time-lapse between arrival to the laboratory and CT scan ranged from 0 days to 24 days (Table 1). Two animals were scanned the same day they were stranded. The other 10 animals were frozen before PMCT scanning (Table 1). Five scanned dolphins not infected with *B. ceti* did not show skeletal or CNS alterations after extensive PMCT analysis (Figure 1). As previously reported for PMCT scans of the CNS of cetaceans [22], the brain midline symmetry, ventricles, and cerebellum were barely resolved, displaying a homogeneous structure in these five *B. ceti* uninfected dolphins. No abnormal focal areas of altered attenuation or signal intensity were observed in the cerebral hemispheres, brainstem, or cerebellum on PMCT scans. Appearance and attenuation of the brain parenchyma and the ventricular system appeared unremarkable, and no evidence of intracranial space-occupying lesion was detected. The skeletons of all five dolphins were unmarked and looked normal. Congruent with this finding, none of these five animals had antibodies against Br-LPS or rendered *Brucella* organisms after the bacteriological examination (Table 1). During necropsy, none of these animals presented gross pathological or histopathological alterations of the CNS, suggesting that stranding occurred for reasons other than infectious disease affecting the brain, including brucellosis. After necropsy, less than 1 mL of CSF was collected from the lateral ventricles. Accordingly, the PMCT images of these five dolphins were used as "normals" for comparison to detect SNC and skeletal abnormalities.

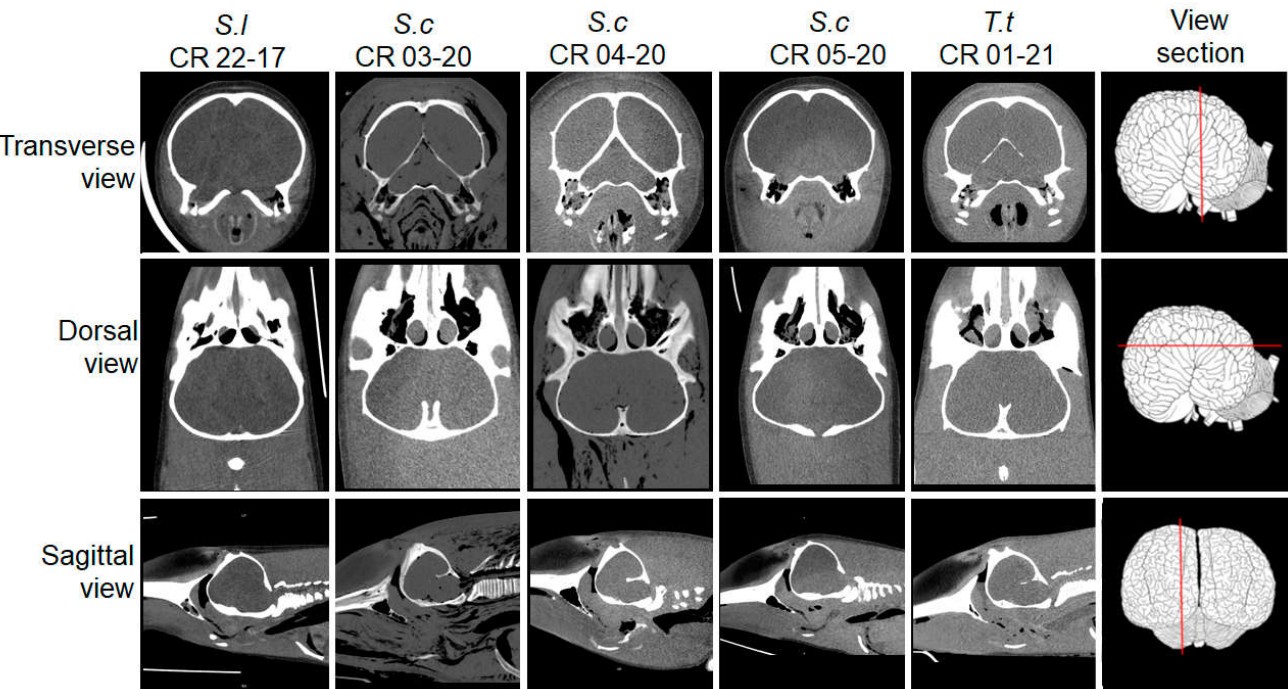

**Figure 1.** Reconstruction by PMCT of transverse, dorsal, and sagittal virtual brain sections of adult dolphins with no signs of CNS alterations. The red line crossing the brain scheme depicted at the far right of the figure indicates the plane of the virtual computed tomography section. *S.l*, *S. longirostris*; *S.c*, *S. coeruleoalba*; *T.t*, *T. truncatus*.

All seven *Brucella*-infected dolphins (Table 1) evaluated by PMCT showed (at least) a bilateral ventriculomegaly based on ventricular length and width of the lateral horns, which was directly associated with hydrocephalus and the amount of CSF collected from the lateral ventricles (Figures 2 and 3). Following this and according to the range of CSF volume collected, we semiquantitatively categorized the PMCT ventriculomegaly as mild (1–1.9 mL of CSF), moderate (2–10 mL of CSF), and severe (>10 mL of CSF), respectively. One juvenile male striped dolphin (CR 03-18) whose ventricular system was barely visible in the transverse, dorsal, and lateral views from the PMCT head images showed a minor ventriculomegaly (Figure 2A). One adult male Fraser's dolphin (CR 09-20) and one adult male striped dolphin (CR 10-20) also showed mild bilateral ventriculomegaly as revealed in the three planes analyzed by PMCT skull images. However, the ventricles were more conspicuous than the CR 03-18 case (Figure 2A). It should be noted that mild ventriculomegaly in neonate cetaceans may be physiologically normal and not necessarily linked to neuropathology. However, the CR 03-18 was a thoroughly developed juvenile dolphin with positive serology and *B. ceti* isolation. Two other adult female striped dolphins (CR 05-19 and CR 07-19) displayed a moderate ventriculomegaly with enlargement of the lateral ventricles and the third ventricle in the reconstruction of transverse, dorsal and lateral images (Figure 2B). Finally, one adult (CR 08-20) and one female calf (CR 01-18) striped dolphins displayed severe ventriculomegaly with significant enlargement of the two lateral ventricles, the third and the fourth ventricles, as demonstrated in the different PMCT scanning planes (Figure 3). In both of these severe cases, the mesencephalic aqueduct could be observed. In six out of seven dolphins displaying ventriculomegaly, *B. ceti* ST 26, genotype P1, was isolated from the CSF and identified as the etiological agent of neurobrucellosis. In contrast, *Brucella* was not isolated from the tissues of Fraser's dolphin, showing mild ventriculomegaly. However, the *Brucella* infection in this dolphin was indicated by antibodies against Br-LPS.

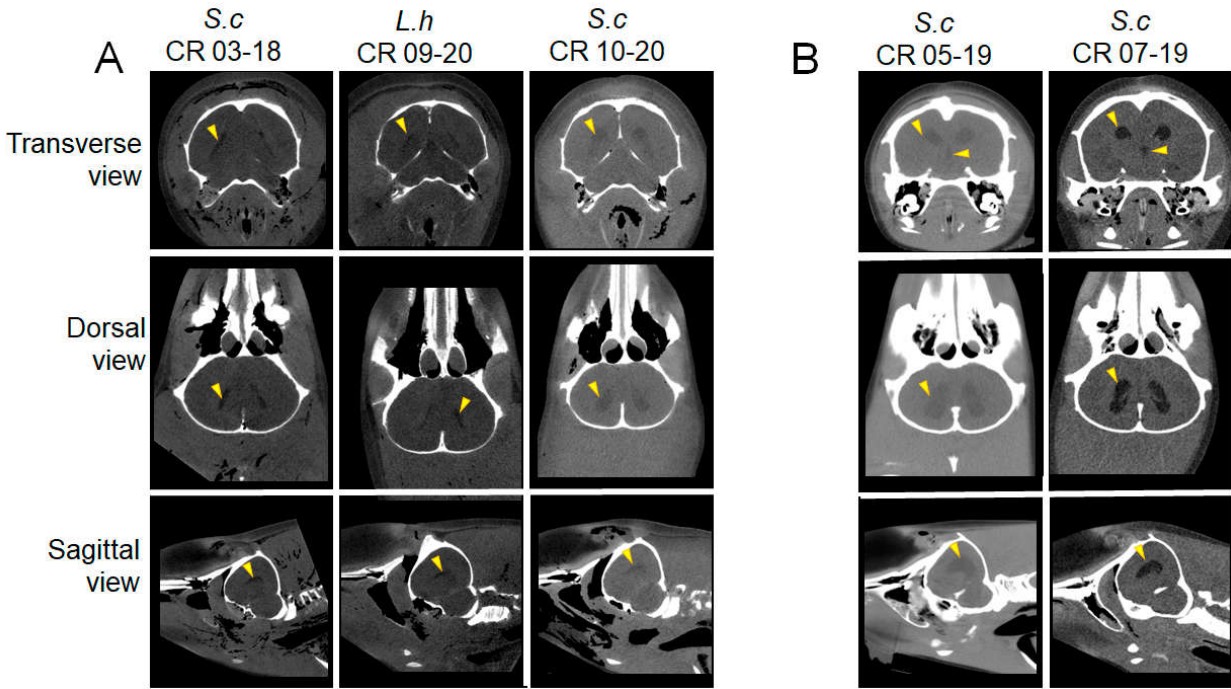

**Figure 2.** Reconstruction by PMCT of transverse, dorsal, and sagittal virtual brain sections of adult dolphins with neurobrucellosis, showing mild ventriculomegaly (yellow point arrows). (**A**) Mild ventriculomegaly. (**B**) Moderate ventriculomegaly. The planes of the virtual computed tomography sections are as presented in Figure 1. *S.c*, *S. coeruleoalba*; *L. h*, *Langenodelphis hosei.*

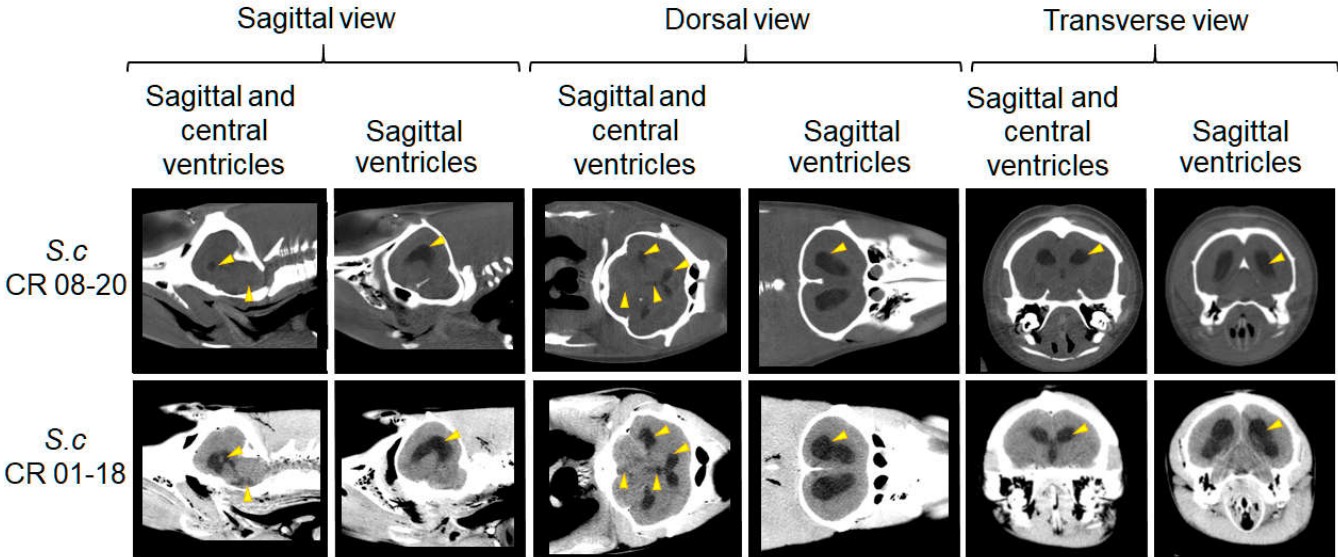

**Figure 3.** Reconstruction by PMCT of sagittal, dorsal, and transversal virtual brain sections of adult dolphins with neurobrucellosis showing severe ventriculomegaly in the two lateral ventricles and the third and fourth ventricles (yellow point arrows). The planes of the virtual computed tomography sections are as presented in Figure 1. *S.c*, *S. coeruleoalba.*

Two male striped dolphins, one adult and one juvenile, showed osteoarticular lesions associated with brucellosis (Figure 4). In the adult animal (CR 10-20), severe fibrinosuppurative osteoarthritis of the atlantooccipital joint was characterized by diffuse roughening of the articular surfaces of occipital condyles and atlas. These structures exhibited

multifocal to coalescing areas of articular cartilage eburnation, necrosis, and severe osteolysis. A focal area of severe bone remodeling and osteolysis observed over the dorsolateral aspect of the left occipital condyle was associated with a focal full-thickness defect in the occipital bone, as confirmed by PMCT (Figure 5b). The tridimensional CT reconstruction of the caudal view of the skull reveals the magnitude of the lesion (Figure 4A). Likewise, in the juvenile striped dolphin (CR 03-18), the caudal vertebrae had hyperattenuating changes at the level of Ca3-Ca4 with intervertebral disc space displaying evident osteolysis of the vertebral endplates, which also becomes evident in the tridimensional CT reconstruction of the spine (Figure 4B). Both lesions likely hampered mobility and swimming performance due to the associated inflammation in these sites. Both animals also presented mild ventriculomegaly and lymphohistiocytic meningoencephalomyelitis.

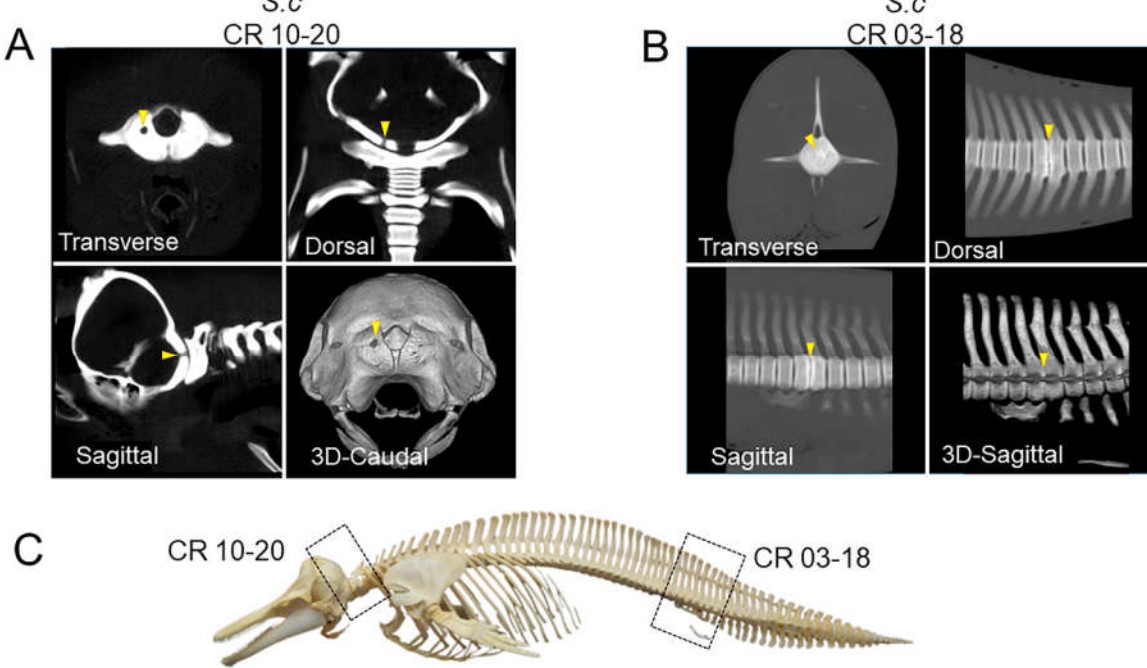

**Figure 4.** Reconstruction by PMCT of virtual cranial and spinal sections of adult dolphins with neurobrucellosis. (**A**) Transverse, dorsal, sagittal, and caudal views of the cranial bones and cervical vertebrae. The caudal view corresponds to a tridimensional reconstruction of the skull occipital region (3D). Yellow arrows indicate osteolysis of the occipital condyle in the different views. (**B**) Transverse, dorsal, and sagittal views of the tail bones showing fusion of Ca3 and Ca4 caudal vertebrae (yellow arrow). The sagittal view corresponds to the tail region's tridimensional reconstruction (3D). (**C**) The planes of the corresponding virtual computed tomography sections are indicated in the punctuated rectangles. *S.c*, *S. coeruleoalba. B. ceti* was isolated from the osteolytic lesion in the occipital condyle. Dolphin skeleton from Sklmsta~commonswiki, Creative Commons, under universal public domain license CC0 1.0.

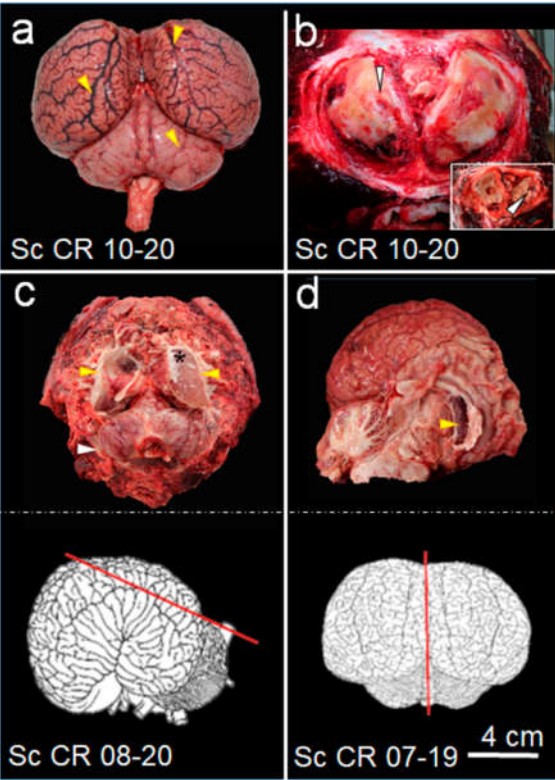

**Figure 5.** The brain and brain sections of *S. coeruleoalba* (S.c) striped dolphins with neurobrucellosis. (**a**) There are mildly thickened and hyperemic meninges over the cerebrum and cerebellum (yellow arrows). (**b**) Caudal view of the skull showing the occipital condyles and the corresponding atlas vertebrae (insert) with osteomyelitis and osteolysis (white arrows). (**c**) Oblique section of a frozen brain showing severe ventriculomegaly (yellow arrows). The right lateral ventricle is filled with frozen cerebrospinal fluid (asterisk); part of the skull bone is depicted (white arrow). The red line crossing the brain scheme, depicted at the bottom of the figure, indicates the plane of the surgical section. (**d**) Sagittal section of the brain with moderate ventriculomegaly, showing the rostral segment of the left lateral ventricle dilated (yellow arrow) of case CR 07-19. The red line crossing the brain scheme, depicted at the bottom of the figure, indicates the plane of the surgical section.

On gross examination, the brains of all 12 animals with positive isolation of *Brucella* from CSF, including 6 striped dolphins displaying bilateral ventriculomegaly on PMCT, showed non-specific changes such as hyperemia, meningeal thickening, and discrete hemorrhages (Figure 5a). Brain extraction and partial dissection were performed on partially frozen specimens to preserve the CSF and achieve bacteriological isolation. Moderate and severe ventriculomegaly observed in the lateral ventricles after PMCT corresponded to the gross pathology of the dissected brain, revealing moderate to severe ventricular dilation with abundant CSF consistent with hydrocephalus (Figure 5c,d).

Histopathology of brain sections of *Brucella* positive culture animals shows classical changes of neurobrucellosis observed in cetaceans [4] characterized by lymphoplasmacytic and histiocytic meningoencephalomyelitis of different severities depending on the individual and the anatomical location within the brain (Figure 6). Histopathological examination did not reveal associated toxoplasma or morbillivirus brain infections. The inflammatory index of different CNS sections of dolphins with neurobrucellosis is presented in Figure 7. Severe inflammatory infiltrates most frequently classified as grade 3 were observed in the meninges and within the perivascular Virchow–Robin spaces of

the spinal cord, medulla oblongata, pons, cerebellum, and less frequently in more rostral locations (Figure 7).

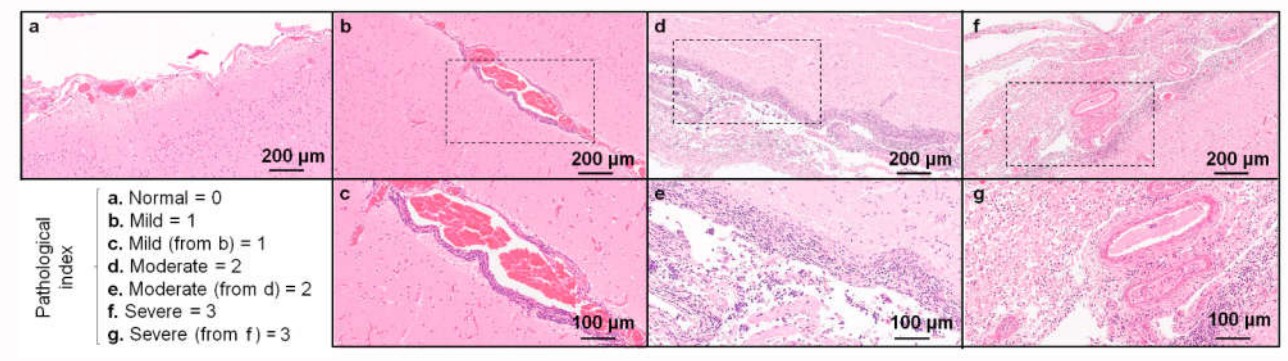

**Figure 6.** Brain histopathology of dolphins with and without neurobrucellosis. (**a**) Normal brain and meninges section with no signs of neurobrucellosis. (**b**) Brain and meninges sections of a *B. ceti* infected dolphin showing mild (**b**,**c**), moderate (**d**,**e**), and severe (**f**,**g**) mononuclear inflammatory infiltrate. The panels "c", "e", and "g" correspond to magnification indicated in the dotted rectangles in "b", "d", and "f," respectively; panel "a" corresponds to case study *S. longirostris* CR 22-17; panels "b" and "c" correspond to case study *S. coeruleoalba* CR 07-19; panels "d" and "e" correspond to case study *S. coeruleoalba* CR 13-17; panels "f" and "g" correspond to *S. coeruleoalba* CR 01-18. Hematoxylin and eosin stain. A scale bar is indicated for each panel.

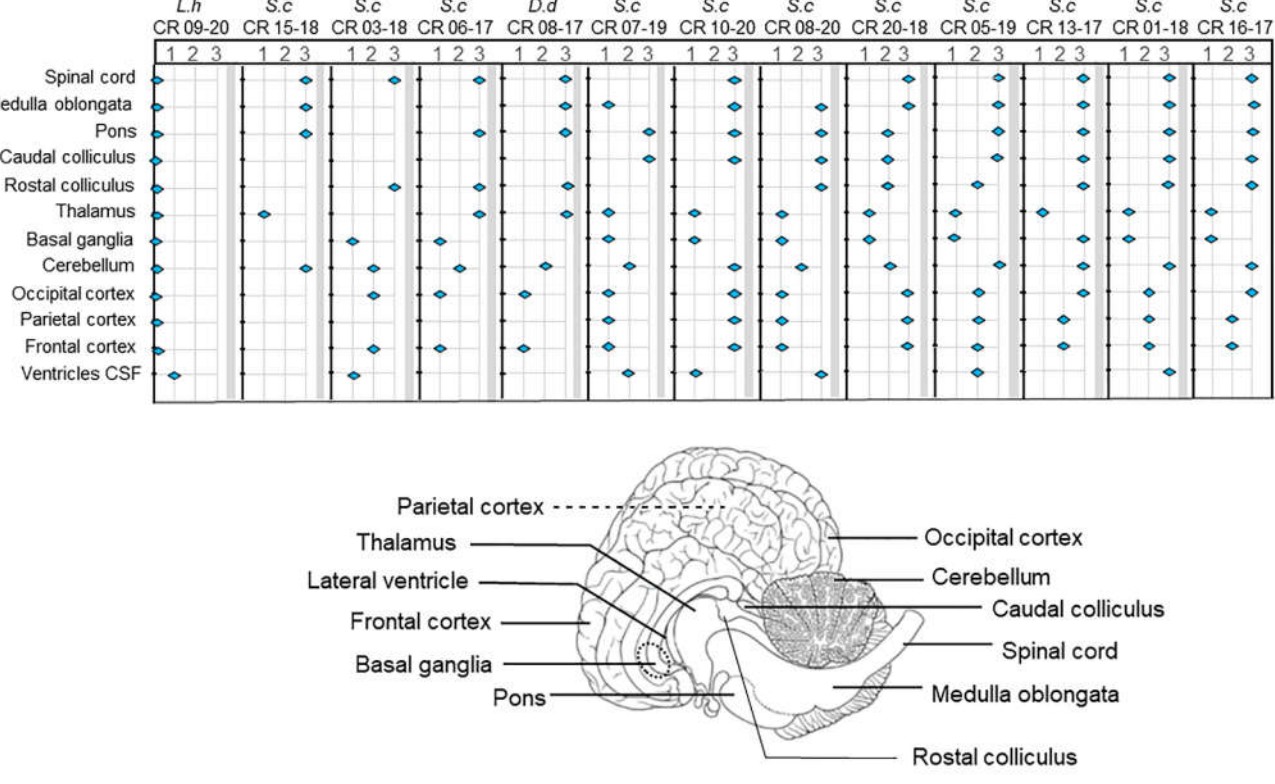

**Figure 7.** Inflammatory index of different central nervous system sections in dolphins with neurobrucellosis. (1) Mild inflammation, (2) moderate inflammation, and (3) severe inflammation. Values below "1" indicate no inflammation. The location of the various brain tissues analyzed is depicted in the lateral brain section scheme. The amount (degree) of CSF in the lateral ventricles was estimated after necropsy of the brain, as shown in Figure 5c.

Although the amount of CSF collected from the brain positively correlated with the different levels of ventriculomegaly estimated in the PMCT scans, the hydrocephalus severity does not strictly correlate with the severity of the inflammatory index determined by histopathological parameters of the brain cortex and other regions of the CNS (Figure 7). These differences suggest that although these processes are linked, the overall neurobrucellosis pathophysiological events that induce brain CNS dysfunction and damage are multifactorial and complex.

The Fraser's dolphin *L. hosei* (CR 09-20), displaying mild ventriculomegaly and positive serology for *Brucella*, was negative for bacteriological isolation from the CSF and did not demonstrate inflammatory changes compatible with neurobrucellosis. This animal only revealed mild lymphocytic infiltrates in the forebrain and incidental Lafora-like polyglucosan bodies in the neuroparenchyma of the frontal cortex.

## 4. Discussion

Similar to neurobrucellosis in humans [16], *Brucella* organisms can invade the CNS of dolphins and cause severe disease. This event is striking since, except for reports in aborted fetuses [28], neurobrucellosis has never been recorded in juvenile or adult bovine, caprine, ovine, swine, or rodent, which are natural hosts of *Brucella* organisms. In mammals, the blood–brain barrier that restricts free access for molecules and pathogens from the blood to the brain seems alike, displaying very similar tight junction complexes between endothelial cells, close contact with pericyte and astrocytes, and low levels of transcytosis within endothelial cells [29]. Nevertheless, humans and dolphins have the highest relative brain size related to body dimensions, with high neuron densities in the cerebral cortex and larger cerebellums among mammals [30]. With their high rate of metabolism, both the human and dolphin brains are actively cooled down by an extensive network that supplies efficient blood flow [31].

Nevertheless, the specific functional properties of the human and dolphin blood–brain barrier remain elusive, and how *Brucella* organisms invade the brain remains unknown. Experiments performed ex vivo have suggested that *Brucella* infected monocytes might cross the blood–brain barrier and serve as vehicles for dispersing *Brucella* organisms causing neurobrucellosis [32]; however, this proposal has not been tested *in vivo*. In addition, an age-dependency leakage of the barrier has been established, increasing with aging in humans [29]. However, as demonstrated here, similar to humans [33], calves, juvenile, and adult dolphins suffer neurobrucellosis, indicating that age is not a decisive factor regarding CNS *Brucella* infections.

The presence of ventriculomegaly diagnosed by CT or MRI in stranded dolphins has been noted before in dolphins infected with *Brucella* organisms, but the CT observations have not been published in the peer-reviewed literature [18,19]. In this respect, it seems that our work is the first peer-reviewed report describing CT scans in animals with proven neurobrucellosis caused by *B. ceti* other than humans. However, a limitation of PMCT scans for diagnosing CNS pathologies is the deterioration of brain tissues as time passes, diminishing the quality of the images [22]. In our case, the prompt transportation and freezing of dolphins in a proper position for less than 2–8 h after the stranding and death were crucial factors for the quality of the PMCT scans and diagnostic reliability. Yet, there are no standard CT parameters to evaluate the degree of ventriculomegaly in dolphins, such as the Evans Index commonly used to estimate hydrocephalus in humans [34,35]. There is one phenomenon in our analysis that should be taken into consideration. Liquid expands close to 9% after the freezing point, a factor that may affect the volume of the CSF and consequently slightly influence the size of the ventricles in the brain of the frozen dolphins analyzed by CT. Moreover, further investigations are needed to estimate, more accurately, the average amounts of CSF in the ventricles of dolphins not infected with Brucella, considering that unremarkable images are common findings in some human patients with neurobrucellosis [16]. Currently, research is in progress to determine these and

other parameters in dolphins for estimating the degree of ventriculomegaly in these animals.

The severity of the hydrocephalus was also assessed by comparative estimation of the amount of CSF present and relative dilatation of the ventricles observed by gross pathology, which positively correlated with the PMCT images depicted as mild, moderate, and severe compared with the unremarkable "normal" brains of dolphins without neurobrucellosis. Accordingly, the diagnosis performed by CT scans was in agreement with the amount of CSF collected and the anatomical examination of the brain ventricles of the dolphins during necropsy.

As in humans with ventriculomegaly due to *Brucella* infections, dolphins also show several neurological problems, such as motor dysfunction, breathing problems, seizures, and loss of consciousness [36,37]. In this respect, our results are commensurate with the CT scans of hydrocephalus recorded in human neurobrucellosis [36,37]. In severe human cases, communicating hydrocephalus recorded by CT scans has been determined as a neurobrucellosis complication [37]. Although anatomically different from humans, we also observed the anatomical communications among the lateral, third, and fourth ventricles in severe hydrocephalus cases in dolphins.

Complications of long-lasting human brucellosis include osteoarticular disorders, mainly in the sacroiliac and spinal joint regions, which are the most commonly affected sites [15]. The osteoarticular destruction may be caused by the direct invasion of *Brucella* or as a result of an ongoing inflammatory response. *Brucella* can infect the osteoblasts and live within them, which can later secrete cytokines and chemokines as pro-inflammatory mediators and metalloproteases that may contribute to the bone and joint destruction observed in patients with osteoarticular complications of brucellosis [38]. *Brucella* spp. can also induce osteoclast activation and cause bone resorption [16]. As other studies have shown [39], we observed discospondylitis in one *Brucella*-infected dolphin, paralleling what has been observed in humans and dolphins before [10]. However, as other authors [14] reported, we also observed skull lesions at the atlantooccipital level in one dolphin and isolated *B. ceti* from this lesion, indicating that this bacterium was the etiological agent responsible for this osteoarticular condition. So far, skull or atlantooccipital osteoarticular lesions are considered extremely rare in humans [16]. As stated, the occipital joint is one of the few synovial joints adapted for movement and weight-bearing stress in dolphins and is therefore prone to be invaded by *Brucella* organisms. In this respect, PMCT helped us detect these osteoarticular disorders and guided us to explore these sites and isolate brucellae at necropsy.

As estimated by the pathological index, the meningoencephalitis severity was not strictly related to the degree of ventriculomegaly observed in the dolphins analyzed. Moreover, clinical symptoms such as imparity to swim, seizures, and disorientation followed by death were common features in most stranded dolphins with neurobrucellosis. Additionally, all these animals presented empty stomachs during necropsy, indicating that the disease impairs their foraging ability. Therefore, it seems that there is a threshold of brain infection that induces disorientation and eventually causes stranding, which may relate to different stages of brain inflammation, some depicted as severe hydrocephalus, but others showing milder forms of ventriculomegaly. Indeed, in humans, CT scans of patients with neurobrucellosis displaying positive clinical findings vary from unremarkable "normal" imaging studies to various imaging abnormalities that reflect either an inflammatory process, a vascular insult, or an immune-mediated syndrome [40]. In this respect, the PMCT, although very useful for detecting osteoarticular lesions and hydrocephalus, cannot substitute the pathological and histopathological findings detected after necropsy. In comparison to MRI, the CT possesses less power for resolving anatomical details in the CNS of dolphins [22]. Intracranial structures and identification of intra-parenchymal lesions, not identifiable on CT scans, may, in some cases, be resolved by MRI. Still, CT scans have better spatial resolution than MRI when suitable protocols are used, which

additionally are faster and less expensive to perform, and therefore practical for postmortem analysis, previous to necropsy.

In this study, we only performed CT on dead dolphins. Still, our findings suggest that this technique is a suitable and reliable method to diagnose ventriculomegaly and associated osteoarticular lesions in live dolphins suspected of having neurobrucellosis. Furthermore, the technique can be extended to other CNS diseases that induce strandings in cetaceans. As in humans, live-stranded dolphins suspected of brucellosis occasionally have been subjected to antibiotic regime procedures that have been shown to diminish the symptoms and resolve the infection with a favorable outcome [36]. In addition, CT scans of the affected dolphins may reveal bone lesions that are not obvious in the general clinical analysis and may impair the swimming ability. However, brucellosis is a transmissible infection and a risk to other animals, a factor that must be considered when keeping or realizing exams in endangered species.

In the absence of reliable animal models to study CNS *Brucella* infections in humans, naturally infected dolphins with neurobrucellosis may serve as an archetype to explore the pathology of the CNS and understand how *Brucella* organisms invade the brain. In addition, noninvasive CT scans associated with fast and trustworthy serological and bacteriological diagnoses may be critical for a therapeutic decision for stranded cetaceans suffering brucellosis, mainly in endangered species with scarce information on their population, such as dwarf sperm whales and Hector's dolphins [9,41].

**Author Contributions:** Conceptualization, R.G.-B., and G.H.-M.; methodology, R.G.-B., G.H.-M., and B.C.-W.K., C.T.-S.; software, A.G.-Z., M.J.R.-M., B.C.-W.K., C.T.-S., and G.H.-M.; validation, R.G.-B., G.H.-M., and B.C.-W.K.; formal analysis R.G.-B., B.C.-W.K., and G.H.-M.; investigation A.G.-Z., M.J.R.-M., R.G.-B., B.C.-W.K., and G.H.-M.; resources, A.G.-Z., R.G.-B., E.B.-C., M.C.-C., and G.H.-M.; data curation A.G.-Z., R.G.-B., E.B.-C., E.M., and G.H.-M.; writing—original draft preparation, E.M., and G.H-M; writing—review and editing, A.G.-Z., R.G.-B., M.J.R.-M., B.C.-W.K., J.D.P.-A., E.B.-C., M.C.-C., M.S.-E., C.G.-V., E.M., and G.H.-M.; visualization R.G.-B. and G.H.-M.; supervision R.G.-B., E.B.-C., and G.H.-M., project administration R.G.-B., E.B.-C., and G.H.-M.; funding acquisition A.G.-Z., R.G.-B., E.B.-C., M.C.-C., and G.H.-M. All authors have read and agreed to the published version of the manuscript.

**Funding:** This research was funded by FOCAES grant UNA-VI-OFIC-76-2020 for AGZ and the National Wildlife Program of the National Animal Health Service (SENASA). Edgardo Moreno received support from the National Academy of Sciences of Costa Rica

**Institutional Review Board Statement:** The study was conducted following the Declaration of Helsinki, approved by the Institutional Review Board of the National Brucellosis Control Program and Wildlife Program of the Costa Rican National Animal Health Service (SENASA), and performed in agreement with the corresponding law "Ley de Bienestar de los Animales" (Ley N° 7451 1994) and to the International Convention for the Protection of Animals endorsed by Costa Rican Veterinary General Law on the SENASA (Ley N°. 8495 2006). All procedures involving live *Brucella* followed the "Reglamento de Bioseguridad de la CCSS 39,975–0, 2012," after the "Decreto Ejecutivo No. 30,965-S," the year 2002 and research protocol 0045–17 approved by the National University, Costa Rica. According to the Biodiversity Law No. 7788 of Costa Rica and the Convention on Biological Diversity, the genetic resources were accessed under the terms regarding an equal and fair distribution of benefits for those who provided resources under CONAGEBIO, Costa Rica, permit No. R-CM-UNA-003–2019-OT-CONAGEBIO.

**Data Availability Statement:** This does not apply.

**Acknowledgments:** We thank all the personnel of the Aquatic Animal Virtopsy Lab, Hong Kong and SENASA of Costa Rica, especially to the Central Pacific Region and LANASEVE, Hellen Porras Fernández, Juan Carlos Alvarado, Osvaldo Barrantes Granados, Jacqueline Cubillo Morera, Josimar Estrella Morales, Eunice Viquez Ruiz, and the personnel from Parque Marino del Pacífico for their collaboration with the localization and handling of the stranded animals. Additionally, we want to give a special thanks to Jimmy Vargas Olivares from the Hospital San Vicente de Paul for assisting in the histopathological preparations. We also thank the Municipal Police and Coastguard of Costa Rica, the personnel from the Program of Investigation of Tropical Diseases (PIET) to Nazareth Ruiz

Villalobos, and Karol Roca for their collaboration during necropsies. This work was supported and approved as part of the National Program of Wildlife of SENASA San José, Costa Rica.

**Conflicts of Interest:** The authors declare no conflict of interest. The funders had no role in the study's design, collection, analyses, or interpretation of the data, in the writing of the manuscript, or in the decision to publish the results.

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
