# Peer review of "Pathological Studies and Postmortem Computed Tomography of Dolphins with Meningoencephalomyelitis and Osteoarthritis Caused by Brucella ceti"

_2673-1924, doi:10.3390/oceans3020014_

Round 1
Reviewer 1 Report
"Pathological studies and postmortem computed tomography of dolphins with meningoencephalomyelitis and osteoarthritis caused by Brucella ceti" is an important manuscript that will be a welcome addition to the available literature for stranded marine mammals. As the authors state PMCT and antemortem CT is used to assess for ventriculomegaly in stranded dolphins but no peer-reviewed manuscript has thus far been published regarding that. There are a few things in the manuscript that need clarification. The main thing is the volumes of CSF described. It is unclear through the manuscript whether the volumes provided as cut offs are quantitative or qualitative and how the volume “estimates” were reached. This is needed as there are several broad statements such as “The severity of ventriculomegaly by PMCT does not directly correlate with the severity of the inflammatory index determined by histopathological parameters of the brain cortex and other CNS regions, suggesting that these processes, although linked, are multifactorial and need further characterization and validation to establish a better cutoff on the PMCT.” Without clarifying how volume was determined this is an invalid statement. In the partly frozen cadavers/brains there is no comment on whether the volume of CSF would be expected to expand, as is typical for most water based liquids, and how that may have affected analyses of the data.
Some specific points are listed below:
Line 27: Change computerized to computed tomography.
Line 29: Central Nervous System should be lower case.
Line 62: Suggest rewording something like “….uncoordinated swimming with lateral listing, buoyancy disturbances, and death.”
Line 111: Replace limbs with fins
Line 112: Change necropsy findings to necropsy examination
Line 124: I think shorelines is supposed to be shorelines?
Line 126: Please clarify that DEAD dolphins (or use cadavers instead) only were transported in hermetically sealed bags. Thank you.
Line 130: Change dolphis with dolphins
Lines 172-182: The specific and generalized reconstruction algorithms used for the images should also be stated. For example: B90 used for soft tissue optimization and H61 for bone optimization or whatever was actually selected and used. This is in the DICOM meta data and easily accessible.
Line 175: Change segments with slices
Line 200: This sentence needs rewording as there are too many negatives for it to read clearly: As previously reported for PMCT scans of the CNS of cetaceans [20], the brain midline symmetry, ventricles, and cerebellum were not clearly resolved, displaying a homogeneous structure in these five dolphins not infected with B. ceti
Line 203: What feature are the authors referring to?
Line 205: Intensity is not a term used for CT so should be removed.
Line 207: Vasculature cannot be accurately assessed without IV contrast medium and the statement ruling out vascular anomalies should be removed.
Line 207: I would suggest changing this to say: The skeletons of all five dolphins were normal.
Line 213: Change After necropsy to During necropsy
Line 215: Change used as normal parameters to used as normals for comparison
Line 220: This statement needs clarification: “Following this and according to the volume of CSF collected, we categorized the PMCT ventriculomegaly as mild (1-1.9 ml of CSF), moderate (2-10 ml of CSF), and severe (>10 ml of CSF), respectively.” How are you estimating 1.9mls of CSF? There seem to be some statements suggesting the CSF is being measured quantitatively then in other places its being subjectively assessed and perhaps both are being performed but it is unclear to the reader and vital that this is understood for many of the conclusions made.
Line 224: Change “PMCT skull images presented”…to “PMCT head images showed…”. The entire head not just the skull was imaged.
Line 274: Change dorso lateral to dorsolateral
Figure 1, 2, 3, 4 legends: Change Reconstruccion to Reconstruction and lateral to sagittal
Figure 3 Legend changes four ventricles to fourth
Images in figure 4: Change transversal to transverse
Figure 5 legend: Change central nervous system to Brain; Change anterior segment of the left lateral ventricle to rostral segment of the left lateral ventricle, in line with the Nomina Anatomica Veterinaria
Line 365: Change to say first peer-reviewed work. The presence of ventriculomegly on CT or MRI in stranded dolphins is widely accepted as a sign for neurobrucellosis based on published pathology work indicating the presence of ventriculomegaly in may cases, but the CT observations have not specifically been published in peer-review literature. You may want to include this reference regarding the osteoarthritis changes/discospondylitis: Ueno et al. 2020 J. Vet. Med. Sci 82(6): 754-758: Molecular characterization of Brucella yeti from a bottlenose dolphin (Tursiops truncates) with osteomyelitis in the western Pacific.
Line 389: This statement needs to be slightly reworded. There is always communication between the ventricles so perhaps change to read something like: We were only able to observe the anatomical communications between the lateral, third, and fourth ventricles in severe hydrocephalus cases in dolphins.
Line 402: Suggest changing wound with lesion
Line 414: Suggest changing hunting with foraging
Lines 423-424: This statement needs clarification. CT typically has better spatial resolution than MRI depending on the protocol used. But contrast resolution is superior in MRI allowing better distinction of intracranial structures and identification of intra-parenchymal lesions that will not be identifiable on survey CT.
Line 430: This statement is a stretch. You state yourself that there are frequently osteoarthritis changes that affect animal mobility. Those alone would render an animal non-releasable if also arguing that they are affecting ability to swim and dive to forage. Discospondylitis is notoriously difficult to resolve. Brucella is a risk to other animals so suggesting such an animal should be kept in a captive-maintained environment with conspecifics is questionable. Use of an external ventricular drainage system in an animal that is essentially aquatic obligate is also questionable. Diminished symptoms will not be adequate to consider a dolphin releasable. I think its fair to state this is used in humans and may be worth considering in the case of endangered species but the cost associated with the ICU care necessary to even contemplate this type of invasive procedure, let alone the ethics, will be prohibitive to the vast majority of international stranding networks. I would suggest rewording the closing paragraphs.
Author Response
Answers to Reviewer 1:
General comment Reviewer #1: "Pathological studies and postmortem computed tomography of dolphins with meningoencephalomyelitis and osteoarthritis caused by Brucella ceti" is an important manuscript that will be a welcome addition to the available literature for stranded marine mammals. As the authors state PMCT and antemortem CT is used to assess for ventriculomegaly in stranded dolphins but no peer-reviewed manuscript has thus far been published regarding that.
Answer: We thank the reviewer for his/her comment
There are a few things in the manuscript that need clarification.
The main thing is the volumes of CSF described. It is unclear through the manuscript whether the volumes provided as cut offs are quantitative or qualitative and how the volume "estimates" were reached. This is needed as there are several broad statements such as "The severity of ventriculomegaly by PMCT does not directly correlate with the severity of the inflammatory index determined by histopathological parameters of the brain cortex and other CNS regions, suggesting that these processes, although linked, are multifactorial and need further characterization and validation to establish a better cutoff on the PMCT." Without clarifying how volume was determined this is an invalid statement. In the partly frozen cadavers/brains there is no comment on whether the volume of CSF would be expected to expand, as is typical for most water based liquids, and how that may have affected analyses of the data semicuantitativy. Some specific points are listed below:
We thank the reviewer for his/her comments and criticisms. The main criticism regarding the CSF volume has been clarified under the revised version's M&M and Results sections line 170-172, 240,241. Specifically, the query has been answered following the query below. Likewise, we have clarified the significance of ventricular "volume" considering the expansion of frozen liquids under the lines 397-400.
Line 27: Change computerized to computed tomography.
Corrected as suggested (See Line 29)
Line 29: Central Nervous System should be lower case.
Corrected as suggested (see Line 31)
Line 62: Suggest rewording something like "….uncoordinated swimming with lateral listing, buoyancy disturbances, and death."
Corrected. (See Lines 62-63 )
Line 111: Replace limbs with fins
Corrected as suggested (see line114 )
Line 112: Change necropsy findings to necropsy examination
Corrected as suggested ( see line114 )
Line 122: I think shorelines is supposed to be shorelines?
We believe the reviewer meant to change the misspelled "litorals" for "littorals" (See line125 )
Line 136: Please clarify that DEAD dolphins (or use cadavers instead) only were transported in hermetically sealed bags. Thank you.
Corrected as suggested (see Line138 )
Line 130: Change dolphis with dolphins
Corrected as suggested (See Line 138)
Lines 172-182: The specific and generalized reconstruction algorithms used for the images should also be stated. For example: B90 used for soft tissue optimization and H61 for bone optimization or whatever was actually selected and used. This is in the DICOM meta data and easily accessible.
Corrected as suggested (See Line152-157)
Line 175: Change segments with slices
Corrected as suggested (Line151)
Line 200: This sentence needs rewording as there are too many negatives for it to read clearly: As previously reported for PMCT scans of the CNS of cetaceans [20], the brain midline symmetry, ventricles, and cerebellum were not clearly resolved, displaying a homogeneous structure in these five dolphins not infected with B. ceti.
As suggested by the reviewer, the sentence was rephrased (See Lines170-172 )
Line 203: What feature are the authors referring to?
The antecedent was included, and the sentence was rephrased, accordingly. (See Line 224-226 )
Line 205: Intensity is not a term used for CT so should be removed.
Corrected as suggested (see Line 223-225)
Line 207: Vasculature cannot be accurately assessed without IV contrast medium and the statement ruling out vascular anomalies should be removed.
Corrected as suggested (see Line 223-225)
Line 207: I would suggest changing this to say: The skeletons of all five dolphins were normal.
Corrected as suggested (see Line 227-228)
Line 213: Change After necropsy to During necropsy
Corrected as suggested (see Line231)
Line 215: Change used as normal parameters to used as normals for comparison
Corrected as indicated (See . Line 235).
Line 220: This statement needs clarification: "Following this and according to the volume of CSF collected, we categorized the PMCT ventriculomegaly as mild (1-1.9 ml of CSF), moderate (2-10 ml of CSF), and severe (>10 ml of CSF), respectively." How are you estimating 1.9mls of CSF? There seem to be some statements suggesting the CSF is being measured quantitatively then in other places its being subjectively assessed and perhaps both are being performed but it is unclear to the reader and vital that this is understood for many of the conclusions made.
The criticism of the reviewer is well taken. We have clarified this under the M&M section (Line170-172) and Results section (Lines 240-241)
Line 224: Change "PMCT skull images presented"…to "PMCT head images showed…". The entire head not just the skull was imaged.
Corrected as suggested (see Line 244)
Line 274: Change dorso lateral to dorsolateral
Corrected as indicated (see Line295- 296)
Figure 1, 2, 3, 4 legends: Change Reconstruccion to Reconstruction and lateral to sagittal
Corrected as indicated (see lines 316,321,325,330)
Figure 3 Legend changes four ventricles to fourth
Corrected as indicated. Line 326)
Images in figure 4: Change transversal to transverse
Corrected as indicated, Line 331.
Figure 5 legend: Change central nervous system to Brain; Change anterior segment of the left lateral ventricle to rostral segment of the left lateral ventricle, in line with the Nomina Anatomica Veterinaria
Corrected as indicated. Line 346
Line 365: Change to say first peer-reviewed work. The presence of ventriculomegly on CT or MRI in stranded dolphins is widely accepted as a sign for neurobrucellosis based on published pathology work indicating the presence of ventriculomegaly in may cases, but the CT observations have not specifically been published in peer-review literature. You may want to include this reference regarding the osteoarthritis changes/discospondylitis: Ueno et al. 2020 J. Vet. Med. Sci 82(6): 754-758: Molecular characterization of Brucella yeti from a bottlenose dolphin (Tursiops truncates) with osteomyelitis in the western Pacific.
The comment is well taken. We have rephrased the sentence accordingly and included appropriate references of non-peer review reports (Congress). (See Lines 386-388)
The suggested reference is not related to neurobrucellosis; though it was included in a different section related to bone lesions (See Lines 427)
Line 389: This statement needs to be slightly reworded. There is always communication between the ventricles so perhaps change to read something like: We were only able to observe the anatomical communications between the lateral, third, and fourth ventricles in severe hydrocephalus cases in dolphins.
As suggested by the reviewer, the sentence was rephrased as suggested (See Line 419)
Line 402: Suggest changing wound with lesion
Corrected as suggested (see Line 432)
Line 414: Suggest changing hunting with foraging
Corrected as suggested (see Line 444)
Lines 423-424: This statement needs clarification. CT typically has better spatial resolution than MRI depending on the protocol used. But contrast resolution is superior in MRI allowing better distinction of intracranial structures and identification of intra-parenchymal lesions that will not be identifiable on survey CT.
The sentence has been rephrased following the reviewer's suggestion (see Lines 454-456)
Line 430: This statement is a stretch. You state yourself that there are frequently osteoarthritis changes that affect animal mobility. Those alone would render an animal non-releasable if also arguing that they are affecting ability to swim and dive to forage. Discospondylitis is notoriously difficult to resolve. Brucella is a risk to other animals so suggesting such an animal should be kept in a captive-maintained environment with conspecifics is questionable. Use of an external ventricular drainage system in an animal that is essentially aquatic obligate is also questionable. Diminished symptoms will not be adequate to consider a dolphin releasable. I think its fair to state this is used in humans and may be worth considering in the case of endangered species but the cost associated with the ICU care necessary to even contemplate this type of invasive procedure, let alone the ethics, will be prohibitive to the vast majority of international stranding networks. I would suggest rewording the closing paragraphs.
We agree with the reviewer's comment; the sentence has been rephrased accordingly (see lines 465-469)
Reviewer 2 Report
- This is a clear, well-written report that should be of interest to many scientists and veterinarians.
- I found no serious errors or problems. I think the authors used appropriate methods, and they describe their methods and findings clearly and effectively. I have few suggestions for improving the manuscript, except for some minor things I list below.
- The manuscript anticipated and answered most of my questions (for example, about transmission of brucellosis from cetaceans to humans, and about the blood-brain barrier). The way the authors present their findings is good, as is the Discussion of their significance.
- One question I still had involves mysticetes (baleen whales). In the first sentence of the Introduction the authors mention the total number of all cetacean species, and they cite two sources that include mysticetes [1 & 2; 2 is specifically about mysticetes]. However, they do not state whether brucellosis is known in Mysticeti, even if their study focused just on dolphins.
- This study looked at 18 dolphins of five species, but it is unclear from the title or early parts of the manuscript if these are all dolphins of one species or of multiple species. In fact, the different species involved are not mentioned until the Results section. I would not wait so long to disclose the species involved. They do not need to be listed in the title, but it seems a bit odd that the authors open the paper by mentioning all of the many species of Costa Rican cetaceans and then don’t mention which specific species they examined until halfway through the paper.
- My main concern involves the cause of death in these different dolphin specimens, and specifically whether the pathological conditions found in this study (e.g., hydrocephalus) might be attributed to some cause other than, or in addition to, brucellosis. Is it possible that the dolphins died and/or stranded due to some other problem? If so, might the symptoms seen via PMCT be attributable to another health issue, including disease or trauma? Was the cause of death determined in each of these cases? If so, this information should be discussed and put in Table 1.
- The title of the manuscript has postmortem as one word; the Abstract (in line 27) has it as two words: post mortem. I don’t care which is used, but please be consistent.
- In line 27 of the Abstract, the word “tomography” seems to be missing (before the abbreviation PMCT).
- Some other long terms that are used repeatedly throughout the manuscript (for example, meningoencephalomyelitis) might also be abbreviated with an acronym.
- The name of the genus Brucella should be italicized in line 30 of the Abstract (in “Brucella-infected”).
- In line 88 (and at least five other places in the manuscript), should atlanto-occipital be hyphenated?
- In line 130, dolphins is misspelled as “dolphis.”
- It is highly unusual to have a separate section just for figures and tables, instead of placing these within the text where they are first mentioned.
- Several figure captions begin with the word Reconstrucción, which should be translated into English (as Reconstruction).
- The figures are clear and easy to follow, with nice images. However, Figure 1 shows “dorsal” and “lateral planes.” These are labeled in other figures as “dorsal” and lateral views,” which is a better way to describe the images. Or the authors could relabel those as frontal and sagittal planes (instead of dorsal and lateral planes), which is how those planes are typically described. In fact, the term sagittal is used in Figure 4 and its caption.
Apart from these very minor issues, the neuro- and clinical anatomy and pathology are all good and are described well.
Author Response
We thank the reviewer for his/her comment
Answers to Reviewer 2:
General comment Reviewer #2: This is a clear, well-written report that should be of interest to many scientists and veterinarians.I found no serious errors or problems. I think the authors used appropriate methods, and they describe their methods and findings clearly and effectively.
Answer: We thank the reviewer for his/her comments. In the following lines, we have answered the specific queries raised by the reviewer
I have few suggestions for improving the manuscript, except for some minor things I list below.
The manuscript anticipated and answered most of my questions (for example, about transmission of brucellosis from cetaceans to humans, and about the blood-brain barrier). The way the authors present their findings is good, as is the Discussion of their significance.
We thank the reviewer for /his/her comments.
One question I still had involves mysticetes (baleen whales). In the first sentence of the Introduction the authors mention the total number of all cetacean species, and they cite two sources that include mysticetes [1 & 2; 2 is specifically about mysticetes]. However, they do not state whether brucellosis is known in Mysticeti, even if their study focused just on dolphins.
The reviewer's comment is well taken. Indeed brucellae have been described in mysticetes like Minke whale ceti; e.g., doi.org/10.7589/2016-08-200 and doi.org/10.3354/dao02986
However, for the moment and in the context of this manuscript, this information is out of the scope
This study looked at 18 dolphins of five species, but it is unclear from the title or early parts of the manuscript if these are all dolphins of one species or of multiple species. In fact, the different species involved are not mentioned until the Results section. I would not wait so long to disclose the species involved. They do not need to be listed in the title, but it seems a bit odd that the authors open the paper by mentioning all of the many species of Costa Rican cetaceans and then don't mention which specific species they examined until halfway through the paper.
We have followed the reviewer's suggestion and included the species studied under the first paragraph of M&M (see Lines 123-124)
My main concern involves the cause of death in these different dolphin specimens, and specifically whether the pathological conditions found in this study (e.g., hydrocephalus) might be attributed to some cause other than, or in addition to, brucellosis. Is it possible that the dolphins died and/or stranded due to some other problem? If so, might the symptoms seen via PMCT be attributable to another health issue, including disease or trauma? Was the cause of death determined in each of these cases? If so, this information should be discussed and put in Table 1.
The reviewer's comment is well taken. We routinely perform differential diagnosis of other pathogens such as morbillivirus and toxoplasmosis, which also cause mononuclear inflammation, but the histopathological examination precludes these infections since they affect the parenchyma of the bran in contrast to brucellosis. Other bacterial infections are ruled out by bacteriological culture, and pathognomonic analysis since most of them are pyogenic with a large number of neutrophils. We have stated this under M&M and Table 1. (see Lines 183-186 ,271,272,311,312)
The title of the manuscript has postmortem as one word; the Abstract (in line 27) has it as two words: post mortem. I don't care which is used, but please be consistent.
Corrected in all sections as suggested (see Lines 29)
In line 27 of the Abstract, the word "tomography" seems to be missing (before the abbreviation PMCT)
Corrected as indicated (see Line 29)
Some other long terms that are used repeatedly throughout the manuscript (for example, meningoencephalomyelitis) might also be abbreviated with an acronym.
The suggestion is well taken. However, the manuscript has already many acronyms, and we do not what to overload the manuscript with more acronyms.
The name of the genus Brucella should be italicized in line 30 of the Abstract (in "Brucella-infected").
Corrected as indicated (see Line 32)
In line 88 (and at least five other places in the manuscript), should atlanto-occipital be hyphenated?
Corrected as suggested (see Lines 90,95, 98, 292, 431,433)
In line 130, dolphins is misspelled as "dolphis."
It was corrected as indicated (see Lines 71)
It is highly unusual to have a separate section just for figures and tables, instead of placing these within the text where they are first mentioned.
The formating of the manuscript follows this style
Several figure captions begin with the word Reconstrucción, which should be translated into English (as Reconstruction).
Corrected as indicated (see Lines 316,321, 325,330)
The figures are clear and easy to follow, with nice images. However, Figure 1 shows "dorsal" and "lateral planes." These are labeled in other figures as "dorsal" and lateral views," which is a better way to describe the images. Or the authors could relabel those as frontal and sagittal planes (instead of dorsal and lateral planes), which is how those planes are typically described. In fact, the term sagittal is used in Figure 4 and its caption.
We have corrected all labels as suggested. See figures 1,2,3,4 .
Apart from these very minor issues, the neuro- and clinical anatomy and pathology are all good and are described well.
Answer: We thank the reviewer for /his/her comments
Round 2
Reviewer 1 Report
The manuscript has greatly benefitted from the edits that the authors have made. There are a few more comments that I have:
Lines 106-108: “Noninvasive imaging techniques such as computed tomography (CT) and magnetic resonance imaging (MRI), commonly used to diagnose neurobrucellosis and associated osteoarticular lesions in humans [16,17], have never been used in confirmed B. ceti infected cetaceans.” This statement is misleading and incorrect. It’s actually used frequently in the US in stranded marine mammal medicine and pathology. But it is true cases haven’t been published. So this should be changed to reflect the data hasn’t been published not left to suggest that the method isn’t used. This is correctly stated in the discussion, but the abstract does not reflect the discussion statement.
Line 158: Please change visualize to observe.
Lines 228-230: If all dolphins were dead at the time of stranding, how were clinical signs for neurobrucellosis determined? If live stranded animals were included, the method of assessment for neurobrucellosis signs are not currently included prior to the results and need to be added.
Lines 453: Intracranial structures and identification of intra-parenchymal le- 453 sions that will not be identifiable on CT scans will be resolved by MRI. This is a modified sentence after first review. It’s not correct. MRI does not see everything. Pathology has to be macroscopic and have proton characteristics that can be picked up on the sequences performed. I would strongly urge the authors to change “will” to “may”.
Figure 5 legend (b): Posterior should be replaced with caudal, per the Nomina Anatomic Veterinaria.
One general comment: I think we need to just be a little bit cautious with statements regarding young cetaceans. It is common to see ventriculomegaly that per this study would be considered mild or moderate in severity in neonates relative to adults. It is common for live-stranded neonates to be CTd as part of their screening as well as blood testing in the US. In cases that have had serial testing and imaging, the neonates remained negative at all times for all tested pathogens including brucella, and didn’t seroconvert, and the ventriculomegaly ‘resolved’ with age. So it may be that a degree of ventriculomegaly is normal in the brain development of neonates.
Author Response
Answers to Reviewer 1:
Lines 106-108: "Noninvasive imaging techniques such as computed tomography (CT) and magnetic resonance imaging (MRI), commonly used to diagnose neurobrucellosis and associated osteoarticular lesions in humans [16,17], have never been used in confirmed B. ceti infected cetaceans." This statement is misleading and incorrect. It's actually used frequently in the US in stranded marine mammal medicine and pathology. But it is true cases haven't been published. So this should be changed to reflect the data hasn't been published not left to suggest that the method isn't used. This is correctly stated in the discussion, but the Abstract does not reflect the discussion statement.
Following the reviewer's comment, we have modified the sentence and included the corresponding non-peer review documents in two sections of the manuscript: the introduction (Lines 108-110) and the Discussion (Lines 389-391). We hesitated and decided unnecessary to include a reference concerning this issue in the Abstract since it is far from the scope and intention of a resume. We acknowledge the reviewer's criticism, and following her/his recommendation, we have agreed to include this in two sections of the manuscript. However, "gray" literature not found in the mainstream of scientific databases are difficult to find and not commensurate with the scientific citation in specialized journals; not only because they were not subjected to peer review, and therefore not exposed to objective evaluation, but because of their appearance in the Web is somewhat unstable.
Line 158: Please change visualize to observe.
Changed as suggested (Lines 160-161)
Lines 228-230: If all dolphins were dead at the time of stranding, how were clinical signs for neurobrucellosis determined? If live stranded animals were included, the method of assessment for neurobrucellosis signs are not currently included prior to the results and need to be added.
The reviewer's comment is correct. The sentence was removed and modified accordingly (line 231)
Lines 453: Intracranial structures and identification of intra-parenchymal lesions that will not be identifiable on CT scans will be resolved by MRI. This is a modified sentence after first review. It's not correct. MRI does not see everything. Pathology has to be macroscopic and have proton characteristics that can be picked up on the sequences performed. I would strongly urge the authors to change "will" to "may".
The sentence was rephrased as suggested (line 457- 458)
Figure 5 legend (b): Posterior should be replaced with caudal, per the Nomina Anatomic Veterinaria.
Modified as suggested
One general comment: I think we need to just be a little bit cautious with statements regarding young cetaceans. It is common to see ventriculomegaly that per this study would be considered mild or moderate in severity in neonates relative to adults. It is common for live-stranded neonates to be CTd as part of their screening as well as blood testing in the US. In cases that have had serial testing and imaging, the neonates remained negative at all times for all tested pathogens including brucella, and didn't seroconvert, and the ventriculomegaly 'resolved' with age. So it may be that a degree of ventriculomegaly is normal in the brain development of neonates.
We thank the reviewer for the insight regarding mild ventriculomegaly in neonates. We have included a short sentence indicating this in the manuscript (lines 249-252). Still, in our case, we did not have neonates but just a juvenile (191cm) that was a fully developed dolphin with positive serology, bacterial isolation, and bone lesions.
Reviewer 2 Report
I thank the authors for carefully considering my comments and for revising the manuscript appropriately. In my view, the revised manuscript is much improved and is now in good shape.
Author Response
I thank the authors for carefully considering my comments and for revising the manuscript appropriately. In my view, the revised manuscript is much improved and is now in good shape
We thank the reviewer's comments.